# Colorectal Cancer Anatomical Site and Sleep Quality

**DOI:** 10.3390/cancers13112578

**Published:** 2021-05-25

**Authors:** Mimi Ton, Nathaniel F. Watson, Arthur Sillah, Rachel C. Malen, Julia D. Labadie, Adriana M. Reedy, Stacey A. Cohen, Andrea N. Burnett-Hartman, Polly A. Newcomb, Amanda I. Phipps

**Affiliations:** 1Public Health Sciences Division, Fred Hutchinson Cancer Research Center, 1100 Fairview Ave N, Seattle, WA 98109, USA; artsil2010@gmail.com (A.S.); rmalen@fredhutch.org (R.C.M.); jlabadie@fredhutch.org (J.D.L.); areedy@fredhutch.org (A.M.R.); Andrea.N.Burnett-Hartman@kp.org (A.N.B.-H.); pnewcomb@fredhutch.org (P.A.N.); aiphipps@uw.edu (A.I.P.); 2Department of Epidemiology, University of Washington School of Public Health, 3980 15th Ave NE, Seattle, WA 98195, USA; 3Department of Neurology, University of Washington School of Medicine, 1959 NE Pacific St, Seattle, WA 98195, USA; nwatson@uw.edu; 4University of Washington Medicine Sleep Center, University of Washington, 908 Jefferson St, Seattle, WA 98104, USA; 5Division of Oncology, University of Washington, 825 Eastlake Ave E, Seattle, WA 98109, USA; shiovitz@uw.edu; 6Clinical Research Division, Fred Hutchinson Cancer Research Center, 1100 Fairview Ave N, Seattle, WA 98109, USA; 7Institute for Health Research, Kaiser Permanente, 2550 S Parker Rd, Aurora, CO 80014, USA

**Keywords:** colorectal cancer, sleep, epidemiology, survivorship

## Abstract

**Simple Summary:**

Around 70% of colorectal cancer patients report having sleeping issues, and identifying whether anatomic site plays a significant factor in sleep quality is important. The aim of our population-based study was to assess differences in sleep between rectal and colon cancer patients. We identified that rectal cancer patients were more likely to have sleep complications, such as changes in sleep patterns after diagnosis, getting up to use the bathroom, and pain, compared to colon cancer patients. Identifying whether anatomic colorectal cancer site affects sleep quality and sleep-related issues suggests that sleep-focused survivorship care may be suggested for rectal cancer patients to ensure patients receive appropriate support.

**Abstract:**

Purpose: Sleep quality in relation to anatomic site among colorectal cancer (CRC) patients is not well understood, though discerning the relationship could contribute to improved survivorship care. Methods: We ascertained sleep quality (Pittsburgh Sleep Quality Index) and other personal characteristics within an ongoing population-based study of CRC patients identified through a cancer registry (*N* = 1453). Differences in sleep quality by CRC site were analyzed using chi-square and ANOVA tests. We used logistic regression to estimate odds ratios (ORs) and 95% confidence intervals (CIs) for the association of tumor site with sleep quality concerns, adjusting for patient attributes and time since diagnosis. Results: Sleeping problems were reported by 70% of CRC patients. Overall, participants with rectal (vs. colon) cancer were more likely (OR (95% CI)) to report general trouble sleeping (1.58 (1.19, 2.10)). Rectal cancer patients were also more likely than colon cancer patients to report changes in sleep patterns after cancer diagnosis (1.38 (1.05, 1.80)), and trouble sleeping specifically due to getting up to use the bathroom (1.53 (1.20, 1.96)) or pain (1.58 (1.15, 2.17)), but were less likely to report trouble sleeping specifically due to issues with breathing/coughing/snoring (0.51 (0.27, 0.99)). Conclusion: Overall, rectal cancer patients were more likely to have sleep complications compared to colon cancer patients. This suggests sleep-focused survivorship care may be adapted according to CRC site to ensure patients receive appropriate support.

## 1. Introduction

Sleep problems and disturbances are prevalent in many cancer patients, occurring in up to 90% of patients [1,2,3]. Cancer patients, after diagnosis or during treatment, appear to experience poorer sleep quality than the general population. Among patients undergoing chemotherapy alone, two-thirds of patients report impaired sleep [1].

Studies demonstrate that observed sleep problems in cancer patients have important consequences with respect to health outcomes and quality of life. In particular, deficits related to sleep quality are associated with high blood pressure, inflammation, and negative impacts on metabolism and endocrine function [4,5,6]. Poor sleep quality and irregular sleep habits are also associated with the development of chronic diseases, including stroke, cancer, diabetes mellitus [7,8], and with overall mortality [9]. Sleep disturbances also affect quality of life and have consequences on work productivity [10], fatigue [11], trouble keeping up with social activities, and mood disturbance [12]. 

Colorectal cancer (CRC) is the third most common cancer among both men and women in the United States [13,14]. Efforts to identify post-diagnostic risk factors are important to improving survival and quality of life after diagnosis. Patients with CRC face many treatment-associated challenges related to bowel control. In addition, as with most cancer patients, they confront general concerns related to cancer and its treatment, such as fatigue, anxiety, pain, nausea, and lower quality of life [15,16]. These cancer-related factors negatively impact sleep quality [17]. Identifying issues contributing to poorer sleep quality among CRC patients may assist in identifying at-risk patients and facilitate improved symptom management. Anatomic site may play a role in sleep disturbances, because patients with rectal cancer tend to have different symptoms and undergo different treatments than those with colon cancer. Therefore, site-specific survivorship care recommendations with regards to sleep may be warranted.

To the best of our knowledge, no studies have previously assessed sleep quality of CRC patients according to the anatomic site of their tumor. We therefore examined whether anatomic site (i.e., rectal vs. colon) was associated with sleep quality in a large prospective study of CRC patients.

## 2. Materials and Methods

### 2.1. Study Population

The Cancer Surveillance System (CSS) is a subset of the National Cancer Institute’s Surveillance, Epidemiology and End Results (SEER) registry. CSS is a population-based cancer registry collecting information on all cancer cases occurring in the 13-county western Washington region [18,19]. Each cancer case and its common personal, tumor, and treatment data elements are identified, confirmed, and collected using the North American Association of Central Cancer Registries (NAACCR) protocols [20]. Among the cancer cases in the CSS registry, we used International Classification of Diseases, Oncology, Version 3 (ICD-O-3) codes to identify patients with CRC: C180, C182-C189, C199, and C209. CRC patients who were diagnosed between April 1, 2016 and December 31, 2018 at the ages of 20–74 were eligible for study recruitment (*N* = 2983). Patients were excluded from study contact due to non-sharable names, research opt-out, or being deceased pre-contact. 

For this study, the CSS program contacted eligible CRC patients via mail about 3-months post-diagnosis to inform them of their potential eligibility for research and to allow them to opt out of research contact. Patients who were alive at the start of their recruitment and who did not opt out of research were approached with an introductory study letter and a follow-up telephone call to assess study eligibility and address questions about their study participation and consent. We were unable to determine eligibility in 339 cases and determined 300 cases to be ineligible due to language barriers, residence outside the study area, or inability to participate due to illness and/or impairment. 

A total of 2345 confirmed eligible CRC patients were identified. Of these, 56 (2%) were deceased before enrollment, 294 (13%) were lost to contact, and 541 (23%) declined to participate. A total of 1454 (62%) consented to participate and were subsequently enrolled in the study. A CONSORT diagram detailing this recruitment flow appears in Appendix A. All participants provided informed consent, and this study was approved by the Institutional Review Board of the Fred Hutchinson Cancer Research Center.

### 2.2. Sleep Quality Assessment

Sleep quality was assessed through baseline questionnaires conducted by telephone interview, online-portal, or paper after diagnosis. We assessed sleep quality using components of the standardized Pittsburgh Sleep Quality Index for the past month [21]. Participants were asked about hours of sleep, reasons related to trouble sleeping, trouble staying awake, sleep quality, snoring, stopping breathing during sleep, and change in sleep behavior. Hours of sleep was defined as actual sleep in a typical 24-h period with responses ranging hourly from 5 to 10 h and more than 10 h. Sleep hours were also categorized into <7, 7–8, and ≥9 h, with 7–8 h representing recommended hours of sleep [22]. Reasons related to trouble sleeping included not falling asleep within 30 min, waking up in the middle of the night or early morning, having to get up to use the bathroom, having pain, and having issues breathing comfortably or coughing or snoring loudly. Trouble staying awake was defined as trouble staying awake while driving, eating meals, or engaging in social activities, with responses measured in frequency (not at all, less than 1 time per week, 1 or 2 times per week, and 3 or more times per week). Trouble staying awake was dichotomized to yes (less than 1 time per week, 1–2 times per week, and 3 or more times per week) and no (not at all). Sleep quality was rated as very good, fairly good, fairly bad, or very bad and then dichotomized into poor sleep quality (fairly bad and very bad) and good sleep quality (fairly good and very good). To assess the utilization of sleep medication, participants were asked if they take any medication at bedtime to help with sleep, with responses measured in frequency (not in the past month, less than once a week, 1 or 2 times a week, and 3 or more times a week). Sleep medication utilization was dichotomized to yes (less than once a week, 1 or 2 times a week, 3 or more times a week) and no (not in the past month). To assess change in sleep behavior, participants were asked if their behavior had changed since diagnosed with cancer (yes, no). 

### 2.3. Tumor Site Ascertainment

Tumor location was obtained for all participants from pathology reports to the CSS registry made by local hospitals. Colon cancer was defined as ICD-O-3 codes C180 and C182–C189, and rectal cancer was defined as codes C199 and C209.

### 2.4. Covariates

The study questionnaire also collected detailed data on known and suspected risk factors for CRC, including smoking history, aspirin and other NSAID use, postmenopausal hormone use by women, family history of CRC, body mass index (BMI), CRC screening and treatment, and other lifestyle and medical factors. Demographics were also obtained, including age at cancer diagnosis, sex, educational attainment, and income level.

### 2.5. Statistical Analysis

Differences in sleep quality by CRC site were evaluated with chi-square tests. This was performed for sleep hours (≤5, 6, 7, 8, 9, 10, >10 h) and binary variables (yes, no) such as problems staying awake, bad sleep quality, sleep medication, not sleeping within 30 min, waking up in the middle of the night, using the bathroom, pain, breathing/coughing/snoring, and no problems sleeping.

We used logistic regression to calculate unadjusted and adjusted odds ratios (ORs) and 95% confidence intervals (CIs) to compare the odds of sleep quality components according to CRC site (rectal vs. colon cancer). Components of sleep quality included categorical sleep amount (<7, 7–8, ≥9 h) and dichotomized problems staying awake, poor sleep quality, sleep medication use, issues relating to trouble sleeping, and change in sleep behavior after diagnosis. All estimates of association were adjusted for a priori-determined confounders of age at diagnosis (years), sex (male, female), BMI (kg/m^2^), education (high school or less, some college, college graduate or higher), stage at diagnosis (localized, regional, distant), and time since diagnosis (months). 

In sensitivity analyses, we examined the association between sleep quality and cancer stage at diagnosis. Specifically, we compared the sleep quality among participants with regional- and distant-stage cancers to participants with local-stage cancer adjusted for age at diagnosis, sex, education, cancer site, and time since diagnosis. We additionally assessed sleep quality components according to CRC site stratified by cancer stage at diagnosis: localized, regional, and distant. We also assessed whether having an ostomy was associated with components of the sleep index. Sensitivity analyses excluded BMI, which provided no differing results, due to convergence issues in models. To assess whether chemoradiation and radiation affected the model, we excluded participants without chemotherapy and radiation data and used logistic regression to assess odds of sleep quality components according to CRC site, and additionally adjusted for chemotherapy (yes, no) and radiation (yes, no). 

All statistical analyses were performed in SAS (Version 9.4, SAS Institute, Inc., Cary, NC, USA). All *p*-values were 2-sided, and a *p*-value of < 0.05 was considered statistically significant.

## 3. Results

Rectal cancer patients were more likely than colon cancer patients to be younger (*p* < 0.001), have lower BMI (*p* = 0.050), and to have an ostomy (*p* < 0.001). They were also more likely than colon cancer patients to have received chemotherapy and have undergone radiation therapy (Table 1).

Among all participants, 70% reported issues sleeping (75% among rectal cancer patients and 68% among colon cancer patients). The most prevalent issues involved waking up to use the bathroom (30%), waking up in the middle of sleep (26%), and unable to sleep within 30 min (18%). Nearly half of participants reported sleeping 7–8 h a night. As shown in Table 2, significant differences were found in sleep quality (*p* = 0.005), problems with sleeping (*p* < 0.001), trouble sleeping due to bathroom use (*p* = 0.002), trouble sleeping due to pain (*p* = 0.005), and changes in sleep behavior after diagnosis (*p* = 0.002) between rectal and colon cancer patients. No differences were found in other sleep quality components, including hours of sleep, as seen in Table 2 and Figure 1.

The associations (reported as OR (95% CIs)) between sleep quality and CRC site are shown in Table 3. In our fully adjusted models, there was an association between reports of problems with trouble sleeping and CRC site (rectal versus colon, 1.58 (1.19, 2.10)). There were also statistically significant associations with trouble sleeping due to various sleep issues. Rectal cancer patients were more likely to have trouble sleeping due to getting up to use the bathroom (1.53 (1.20, 1.96)) and pain (1.58 (1.15, 2.17)). However, they were less likely to have trouble sleeping due to issues surrounding breathing, coughing, and/or snoring (0.51 (0.27, 0.99)). Rectal cancer patients were also more likely to experience a change in sleep behavior after cancer diagnosis (1.38 (1.05, 1.80)). No statistically significant associations with CRC site were observed for other metrics of sleep quality, including amount of sleep, problems staying awake, sleep quality, sleep medication, and trouble sleeping due to being unable to fall asleep within 30 min and waking up in the middle of sleep.

Sleep quality and cancer stage analyses are summarized in Appendix A. Patients with regional- (2.27 (1.62, 3.18)) or distant (2.32 (1.54, 3.51))-stage cancer were over twice as likely to sleep 9 or more hours compared to patients with localized-stage cancer. Patients diagnosed at a later stage were also more likely to report taking sleep medication (distant versus localized, 2.13 (1.38, 3.27)) and to report having changes in sleep behavior (distant versus localized, 2.50 (1.73, 3.62)). Distant cancer stage patients were also more likely to have trouble falling asleep within 30 min compared to localized cancer stage patients (1.81 (1.25, 2.62)). No statistically significant associations of CRC stage at diagnosis and other components of sleep quality were seen. When observing the association between sleep quality components and CRC site stratified by cancer stage, among localized-stage cancer patients, rectal cancer cases were more likely to report poor sleep quality (1.75 (1.10, 2.79)), problems with trouble sleeping (1.87 (1.17, 3.00)), and change in sleep behavior after diagnosis (1.91 (1.12, 3.23)) compared to colon cancer patients (Appendix A). Rectal cancer patients were also more likely to report trouble sleeping due to using the bathroom compared to colon cancer patients among localized- (1.66 (1.09, 2.51)) and distant (2.10 (1.16, 3.81))-stage cancer patients. No statistically significant associations were observed between CRC site and sleep components in other stages of cancer.

Among patients with data on chemotherapy and radiation (*N* = 1148), adjusting for chemotherapy and radiation provided similar results for associations between sleep quality components and CRC sites (Appendix A). However, after adjusting for treatment type, no association was found between CRC site and having trouble sleeping due to breathing, coughing, or snoring (0.66 (0.27, 1.63)) and changes in sleep behavior after diagnosis (1.18 (0.79, 1.77)). Associations between other sleep components and CRC site remained not statistically significant. 

No statistically significant associations were observed between ostomy and sleep quality components (data not shown). Adding ostomy to the adjusted model also did not significantly change results.

## 4. Discussion

### 4.1. Main Findings

In this population-based study, we observed statistically significant associations between sleep quality and rectal versus colon cancer site. There was evidence that rectal cancer patients had worse sleep quality than patients diagnosed with colon cancer. To our knowledge, this is the first study to examine differences in sleep quality in cancer patients by CRC anatomic site.

### 4.2. Interpretation of Findings

The study results indicate that sleep quality was worse for patients diagnosed with rectal cancer compared to patients with colon cancer, even though they reported reduced symptoms of sleep-disordered breathing (breathing, coughing, snoring). In particular, they experienced more problems sleeping, having to use the bathroom at night, and pain. Our study also indicates that rectal cancer patients were more likely than colon cancer patients to have changes in sleep behavior after diagnosis. Rectal cancer patients may have worse sleep quality due to distinctive symptoms and different treatments. Standard-of-care treatment differs between the two anatomic sites, with most localized rectal cancer patients requiring neoadjuvant radiation in addition to the chemotherapy that is used for colon cancer patients [23]. Additionally, primary rectal cancer surgery often requires a low anterior resection, which can result in a spectrum of symptoms known as Low Anterior Resection Syndrome, or a permanent colostomy [24]. Low Anterior Resection Syndrome is a collection of symptoms or issues after removing a part of the rectum, including fecal incontinence and evacuation difficulties, that negatively affects quality of life [24]. Symptoms from both Low Anterior Resection Syndrome and colostomies can explain deficits in sleep quality.

Sensitivity analyses also found that later-stage (regional and distant) CRC patients are twice as likely to sleep 9 or more hours per night. Longer duration of sleep has been found to be associated with all-cause mortality [25]. In fact, a statistically significant relationship between sleep duration, including both shorter and longer sleep durations, and survival has been shown [26]. Sleep quality also differed between rectal and colon cancer patients by stage, especially for issues surrounding problems sleeping. 

### 4.3. Implications for Research and Practice

Our study suggests that clinical providers should pay particularly close attention to sleep issues in rectal cancer patients. Given that sleep issues may have implications for recovery and long-term prognosis, the burden of sleep problems among rectal cancer patients in particular presents an area of need. Healthy sleep promotes a healthy immune system, which is important in the cancer continuum, including mortality, recurrence, or second primaries [27]. Studies have shown that sleep quality could negatively impact health. A study with the Behavioral Risk Factor Surveillance System (BRFSS) found that insufficient sleep was significantly associated with diabetes mellitus, coronary heart disease, stroke, high blood pressure, asthma, and arthritis [8]. Similarly, studies from the National Health and Nutrition Examine Survey (NHANES) have indicated that duration and quality of sleep were associated with pre-diabetes and decreased odds of ideal cardiovascular health [28,29]. Compared to the general population, both rectal and colon cancer patients experience a higher prevalence of sleep problems than that of the general population. Data from the WHO’s World Health Survey found that the prevalence of sleep problems was 8%, while our study indicated a prevalence of 70% among CRC patients, and up to 75% among rectal cancer patients [30].

There are multiple hypotheses surrounding sleep deficits and their effects on health. Genes that are implicated in circadian rhythm are also involved with DNA repair, as seen in rhythm/biological clock-associated proteins being involved in tumorigenesis and cancer progression through their relationship with cell cycle and DNA damage response pathways [31]. This may mean that sleep deficits can play a large role in chronic diseases affected by biological clock-associated proteins. There also may be an effect due to the pineal hormone melatonin, which is involved in circadian regulation and facilitation of sleep, inhibition of cancer growth, and enhancement of immune function [27,32]. However, there are also effects of sleep and the immune system that are independent of melatonin, indicating that sleep disruption could lead to immune suppression and lead to predominance in cancer-stimulatory cytokines [32]. A weakened immune system could ultimately impact CRC survivorship, negatively affecting the ability to fight a CRC diagnosis.

The relationship of sleep quality with outcomes within this population is complex. A systematic review of health care needs of cancer survivors reported that these involved psychosocial needs, help with medical issues, and information on cancer, recovery, late treatment effects, and adjusting to life after treatment [33]. An evaluation of adult survivorship programs in 25 US cancer centers found that 56% of centers reported screening less than 25% of survivors for sleep disorders, with few clinicians being well-prepared to conduct a proper sleep evaluation [34]. These prior studies indicate that clinicians may not focus as much on sleep patterns and sleep quality as a cancer survivorship issue.

### 4.4. Strengths and Limitations

The strengths of this study include the large, well-characterized study population and the ability to assess multiple components of sleep. There are some limitations that should be considered in interpreting our results. Our questionnaire elicited details on sleep components in the preceding month. However, more detailed information in terms of a full sleep quality scale, such as the Patient-Reported Outcomes Measurement Information System Sleep Disturbance and Sleep-Related Impairment tool [35] or the full Pittsburgh Sleep Quality Index, could provide more insight and comparable scoring. In addition, we lack objective measures of sleep disturbance, such as polysomnography or consumer sleep technologies [36]. We also only had sleep quality data from one timepoint, so we could not assess the impact of CRC site on sleep over time. The study sample size also limited our analysis on how CRC site may affect an individual sleep hours breakdown. Although our models have accounted for numerous potential confounders, residual confounding may still be present. We also lacked more detailed data on treatment (e.g., type of surgery, type of chemotherapy) and cancer symptoms that could potentially explain the effects of CRC site on sleep quality. Our study was also a case-only analysis, which prevents us from extrapolating these results to indicate an increased risk of CRC associated with aspects of sleep.

## 5. Conclusions

Colon and rectal cancer were differentially associated with sleep quality in this study. Clinical providers that work with cancer survivors, in particular rectal cancer patients, should pay close attention to sleep issues that may adversely impact health outcomes among these cancer survivors. Additional studies should explore the driving factors behind the association between CRC anatomic site and sleep quality to better address clinical evaluation and management of sleep problems among CRC survivors.

## Figures and Tables

**Figure 1 cancers-13-02578-f001:**
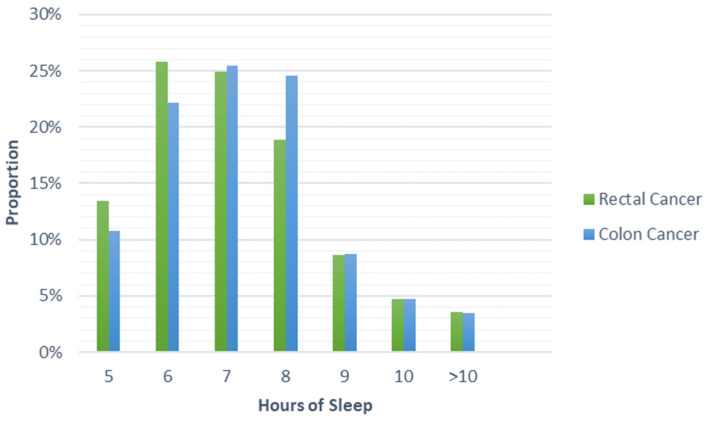
Proportion of cases reporting hours of sleep by colorectal cancer site.

**Table 1 cancers-13-02578-t001:** Baseline characteristics of colorectal cancer patients by cancer site (*N* = 1453).

	Overall *N* = 1453 *N* (%)	Rectal Cancer *N* = 543 *N* (%)	Colon Cancer *N* = 910 *N* (%)	*p*-Value
Age (years)	57.6 ± 10.6	55.6 ± 10.2	58.8 ± 10.7	<0.001
Sex				0.270
Male	789 (54.3)	305 (56.2)	484 (53.2)	
Female	664 (45.7)	238 (43.8)	426 (46.8)	
Race				0.127
Caucasian/White	1144 (78.7)	416 (76.6)	728 (80.0)	
Person of Color ^1^	252 (17.3)	108 (19.9)	144 (15.8)	
Missing	57 (3.9)	19 (3.5)	38 (4.2)	
Education				0.173
Less than HS or HS graduate ^2^	295 (20.3)	124 (22.8)	171 (18.8)	
Some college	494 (34.0)	183 (33.7)	311 (34.2)	
College graduate or higher	650 (44.7)	229 (42.2)	421 (46.3)	
BMI				0.050
<25	526 (36.2)	208 (38.3)	318 (35.0)	
25–30	444 (30.6)	178 (32.8)	266 (29.2)	
>30	446 (30.7)	143 (26.3)	303 (33.3)	
Missing	37 (2.6)	14 (2.6)	23 (2.5)	
Income				<0.001
Less than $30,000	316 (21.8)	134 (24.7)	182 (20.0)	
$30,000–$69,000	346 (23.8)	99 (18.2)	247 (27.1)	
$70,000+	682 (46.9)	276 (50.8)	406 (44.6)	
Missing	109 (7.5)	34 (6.3)	75 (8.2)	
Stage				0.004
Localized	531 (36.6)	173 (31.9)	358 (39.3)	
Regional	622 (42.8)	261 (48.1)	361 (39.7)	
Distant	267 (18.4)	93 (17.1)	174 (19.1)	
Missing	33 (2.3)	16 (3.0)	17 (1.9)	
Ostomy				<0.001
Yes	290 (20.0)	210 (38.7)	80 (8.8)	
No	1157 (79.6)	330 (60.8)	827 (90.9)	
Chemotherapy ^3^	647 (44.5)	254 (46.8)	393 (43.2)	<0.001
Radiation ^3^	185 (12.7)	170 (31.3)	15 (1.7)	<0.001
Time Since Diagnosis (Months)	6.9 ± 3.5	7.3 ± 3.8	6.7 ± 3.4	0.001

NOTE: Some percentages do not sum to 100 due to missing data and rounding. ^1^ Includes African American/Black, Latino, Hispanic, or Spanish Origin, American Indian/Alaska Native, Asian, Native Hawaiian or other Pacific Islander, Multiethnic. ^2^ HS: high school. ^3^ Missing for > 20% of all participants.

**Table 2 cancers-13-02578-t002:** Differences in sleep quality by CRC site (N = 1453).

	Overall *N* = 1453 *N* (%)	Rectal Cancer *N* = 543 *N* (%)	Colon Cancer *N* = 910 *N* (%)	*p*-Value
Sleep Duration				0.244
≤5 h	166 (11.4)	71 (13.1)	95 (10.4)	
6 h	333 (22.9)	137 (25.2)	196 (21.5)	
7 h	357 (24.6)	132 (24.3)	225 (24.7)	
8 h	317 (21.8)	100 (18.4)	217 (23.9)	
9 h	123 (8.5)	46 (8.5)	77 (8.5)	
>10 h	67 (4.6)	25 (4.6)	42 (4.6)	
>10 h	50 (3.4)	19 (3.5)	31 (3.4)	
Problems Staying Awake	195 (13.4)	72 (13.3)	123 (13.5)	0.612
Poor Sleep Quality ^1^	321 (22.1)	141 (26.0)	180 (19.8)	0.005
Sleep Medication Use	259 (17.8)	101 (18.6)	158 (17.4)	0.820
Trouble Sleeping Issues				
Any problems	1019 (70.1)	405 (74.6)	614 (67.5)	<0.001
No sleep within 30 min	261 (18.0)	107 (19.7)	154 (16.9)	0.128
Wake up in middle of sleep	381 (26.2)	152 (28.0)	229 (25.2)	0.147
Use the bathroom	438 (30.1)	189 (34.8)	249 (27.4)	0.002
Pain	209 (14.4)	96 (17.7)	113 (12.4)	0.005
Breathing/cough/snore	54 (3.7)	14 (2.6)	40 (4.4)	0.097
Change in Sleep	268 (18.4)	125 (23.0)	143 (15.7)	0.002

NOTE: Some percentages do not sum to 100 due to missing data and rounding. ^1^ Participants noted their sleep as either “fairly bad” or “very bad”.

**Table 3 cancers-13-02578-t003:** Association of sleep quality among rectal cancer patients compared to colon cancer patients (N = 1453).

	Unadjusted OR (95% CI)	Adjusted ^1^ OR (95% CI)
Sleep Duration		
<7 h	1.31 (1.05, 1.64)	1.22 (0.97, 1.55)
7–8 h	0.78 (0.63, 0.96)	0.84 (0.67, 1.05)
≥9 h	1.00 (0.75, 1.33)	0.99 (0.73, 1.34)
Problems Staying Awake	0.97 (0.71, 1.33)	0.87 (0.63, 1.21)
Poor Sleep Quality	1.40 (1.09, 1.80)	1.26 (0.97, 1.64)
Sleep Medication Use	1.07 (0.79, 1.46)	1.08 (0.78, 1.49)
Trouble Sleeping Issues		
Any problems	1.71 (1.30, 2.25)	1.58 (1.19, 2.10)
No sleep within 30 min	1.24 (0.94, 1.64)	1.13 (0.84, 1.51)
Wake up in middle of sleep	1.20 (0.94, 1.53)	1.15 (0.89, 1.49)
Use the bathroom	1.48 (1.17, 1.87)	1.53 (1.20, 1.96)
Pain	1.56 (1.16, 2.11)	1.58 (1.15, 2.17)
Breathing/cough/snore	0.59 (0.32, 1.09)	0.51 (0.27, 0.99)
Change in Sleep	1.53 (1.18, 1.97)	1.38 (1.05, 1.80)

^1^ Adjusted for age at diagnosis (years), sex (male, female), BMI (kg/m^2^), education (high school or less, some college, college graduate or higher), cancer stage at diagnosis (localized, regional, distant), and time since diagnosis (months).

## Data Availability

The data are publicly accessible to qualified investigators upon request.

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
