# Peer review of "Colorectal Cancer Anatomical Site and Sleep Quality"

_cancers, 2021, doi:10.3390/cancers13112578_

Round 1

Reviewer 1 Report

Comments for authors

Ton et al present the results of sleep quality assessment comparing patients with colon cancer and patients with rectal cancer.

Major comments:

The mentioned results in the paragraph 194-206 do not seem to correspond with the results in table 3. (See more in detail below) These results are also in the abstract. Revisions to the text may be needed when the correct results are mentioned in results and abstract.

Methods:

  • Line 100: I suggest to start a new sentence when confirming the number of patients enrolled in the study. Before, the authors described the groups excluded.
  • The authors do not mention if information regarding cancer treatment was obtained. Please add this to the methods section.

Results:

  • Table 1: the authors state that certain characteristics are more likely to be present in one cancer group compared to the other cancer group (lines 167-170), however no p-values are present in table 1. Please add p-values to table and text.
  • The authors note in the text (line 195-196) that the OR and 95% CI for the association between problems with trouble sleeping and CRC site is 1.58 (1.19, 2.10). In table 3 the mentioned OR is 1.53 (1.20, 1.96). I note the same problem for the OR mentioned in line 199, 201 and 202, and the corresponding topics in table 3. Please clarify/ revise.

In the text, the authors mention that trouble sleeping due to issues surrounding breathing, coughing, and/ or snoring is less likely in rectal patients however the adjusted OR noted in table 3 shows no significance. The same goes for change in sleep behavior after cancer diagnosis.

The mentioned results in the paragraph 194-206 do not seem to correspond with the results in table 3.

  • Please add OR in line 236 and 237
  • Line 240: instead of mentioning (data not shown), I suggest to add the OR and 95% CI here.

Discussion

  • The authors mention the “Low Anterior Resection Syndrome” in line 259 but for readers unfamiliar with the topic, I suggest to elaborate on this syndrome. Would this be a possible explanation for worse sleep and why? Did the authors ask about symptoms known to be associated with this syndrome when interviewing the study participants? It feels as if this is an important topic which needs more details in the discussion.

Author Response

Thank you for your comments. Please see the attachment for our response.

Reviewer 2 Report

Reading this work has given me enormous satisfaction.
In my opinion it is a work of extraordinary originality. The working hypothesis is very novel. The experimental approach is very appropriate for what you are trying to achieve. The results are clear and excellently presented and the discussion is written with absolute clarity for everyone who reads the paper.
Therefore, my recommendation is to accept this work in its current form.

Author Response

(The authors gave the same response as above.)

Reviewer 3 Report

This manuscript is an original article that investigated difference in sleep between rectal and colon cancer patients extracted through a cancer registry in western Washington region. The authors found that 70% of colorectal cancer patients have sleeping problem. In addition, the authors have shown that rectal cancer patients are more likely to have sleep complications compared to colon cancer patients.

This study was conducted well, and the methods are appropriate. The data are presented clearly. In general, this is a well-written paper that presents unique and interesting data. The results will be of interest to clinicians in the field.

However, the following issues need to be addressed:

  1. (Table 1) Please add a P-value to compare the characteristic between colon and rectal cancer. Please describe the Results section regarding Table 1 based on the results of statistical analyses.
  2. (Table 1) Please describe the meaning of “HS”.
  3. (P6L180) “Over half of participants” seems incorrect as only 46% of participants slept 7-8 hours according to Table 2. Please check the results.
  4. (P7L194-206) The descriptions regarding Table 3 do not correspond with the results in Table 3. Please check these data.
  5. The authors stated that sleep problems are associated with overall mortality. I recommend that the authors investigate the relationship between sleep problems and mortality or recurrence in patients with colorectal cancer in the next step.

Author Response

(The authors gave the same response as above.)

Reviewer 4 Report

1.The impact of treatment for colorectal cancer on sleep should also be considered. The methods of treatment are unclear.  The quality of life (QOL) after treatment is quite different depending on endoscopic therapy, surgery, or chemotherapy.

 2. Rectal surgery includes high anterior resection, low anterior resection,  Intersphincteric resection, and Mile's operation. QOL is quite different depending on these operative methods. Do you have these data?

3 Chemotherapy includes adjuvant chemotherapy after surgery, and chemotherapy for stage 4 cancer. Do you have the details? 

Author Response

(The authors gave the same response as above.)

Round 2

Reviewer 1 Report

Thank you for addressing the comments.

Reviewer 4 Report

I have no comment.